# Mineralocorticoid Receptor Antagonists for Preventing Chronic Kidney Disease Progression: Current Evidence and Future Challenges

**DOI:** 10.3390/ijms24097719

**Published:** 2023-04-23

**Authors:** Wataru Fujii, Shigeru Shibata

**Affiliations:** Division of Nephrology, Department of Internal Medicine, Graduate School of Medicine, Teikyo University, Tokyo 173-8605, Japan

**Keywords:** CKD, mineralocorticoids, glomerular filtration rate, albuminuria, podocyte, fibrosis, vasculature

## Abstract

Regulation and action of the mineralocorticoid receptor (MR) have been the focus of intensive research over the past 80 years. Genetic and physiological/biochemical analysis revealed how MR and the steroid hormone aldosterone integrate the responses of distinct tubular cells in the face of environmental perturbations and how their dysregulation compromises fluid homeostasis. In addition to these roles, the accumulation of data also provided unequivocal evidence that MR is involved in the pathophysiology of kidney diseases. Experimental studies delineated the diverse pathological consequences of MR overactivity and uncovered the multiple mechanisms that result in enhanced MR signaling. In parallel, clinical studies consistently demonstrated that MR blockade reduces albuminuria in patients with chronic kidney disease. Moreover, recent large-scale clinical studies using finerenone have provided evidence that the non-steroidal MR antagonist can retard the kidney disease progression in diabetic patients. In this article, we review experimental data demonstrating the critical importance of MR in mediating renal injury as well as clinical studies providing evidence on the renoprotective effects of MR blockade. We also discuss areas of future investigation, which include the benefit of non-steroidal MR antagonists in non-diabetic kidney disease patients, the identification of surrogate markers for MR signaling in the kidney, and the search for key downstream mediators whereby MR blockade confers renoprotection. Insights into these questions would help maximize the benefit of MR blockade in subjects with kidney diseases.

## 1. Introduction: Overview of MR Research

The physiological and pathological roles of the mineralocorticoid receptor (MR) have been intensively studied over the past 80 years, even before the isolation of the receptor itself as well as the endogenous ligand aldosterone. The successful synthesis of 21-hydroxyprogesterone, also known as deoxycorticosterone, by Nobel laureate Reichstein and colleagues in 1937 has opened the way to characterize the actions of “mineralocorticoids” [1], the term introduced by Selye and Jensen [2]. Studies using deoxycorticosterone acetate (DOCA) have shown that this synthetic compound produces potent salt (NaCl) retention in experimental animals [3] and restores plasma volume in patients with Addison’s disease [4,5,6]. Although it is important to note that the biological action of DOCA is not equivalent to that of aldosterone [7], it was also reported that a high dose of DOCA produces toxic effects both in humans and in experimental animals, particularly in the presence of excessive salt [8,9,10].

In 1952, a hitherto unknown compound with potent mineralocorticoid activity was isolated from beef adrenal extract [11], which was subsequently crystallized and named aldosterone by Simpson, Tait, and colleagues [12,13]. The exciting finding was soon followed by the identification of the first case of primary aldosteronism (PA) due to an aldosterone-producing adenoma by Conn [14,15]. The discovery of aldosterone has also facilitated the identification of distinct hypertensive syndromes with low, normal, and high aldosterone levels [16,17,18]. Several steroidal antagonists of aldosterone were rapidly developed, among which was the compound SC-9420, spironolactone [19,20,21,22]. 

With advancements in cloning and sequencing methodology, the gene encoding MR (*NR3C2*) was cloned by Evans and colleagues in 1987 [23]. The primary structure of the major downstream target epithelial sodium channel (ENaC) was also determined [24,25]. Concurrently, the causative genes for the rare Mendelian forms of blood pressure variation were identified mainly by Lifton and colleagues [26]. A series of genetic studies has uncovered that the gain-of-function mutations in the genes encoding aldosterone synthase, MR, and ENaC uniformly drive blood pressure to high levels [27,28,29], whereas the loss-of-function mutations in the same genes all cause hypotensive disorders [30,31,32,33,34], highlighting the predominant role of the aldosterone/MR system in regulating blood pressure in humans [26]. More recently, whole exome sequencing has led to the identification of mutations in *KCNJ5* [35] and other important genes implicated in aldosterone biosynthesis in the zona glomerulosa cells of the adrenal gland, resulting in autonomous aldosterone production [36,37,38,39,40,41,42].

In addition to the critical role of aldosterone and MR in fluid and electrolyte homeostasis, a major interest in the field of chronic kidney disease (CKD) and cardiovascular disease research has been the deleterious effects of MR on renal function and the circulatory system. Indeed, the benefit of MR blockade in patients with chronic heart failure and reduced ejection fraction was demonstrated in multiple clinical studies, which has become the basis for the recommendation of the use of MR antagonists in chronic heart failure [43,44,45]. In the kidney, although a number of experimental studies demonstrated the renoprotective effects of MR blockade in various animal models, it remained to be established in humans whether MR antagonists could actually retard the deterioration of kidney function. A major breakthrough in the field was the development of non-steroidal MR antagonists, which greatly facilitated the basic as well as clinical research regarding the benefit of MR blockade. 

In the following sections, we review the findings from basic science and translational research regarding the physiological and pathophysiological roles of MR in the kidney. We also summarized findings from the clinical studies on the protective effects of MR blockade in CKD. 

## 2. Physiological and Pathophysiological Roles of MR in the Kidney

### 2.1. Electrolyte Handling in Renal Tubules

To adapt to the profound perturbations in the terrestrial environment, our ancestors developed a sophisticated regulatory system that minimizes the changes in the body fluid composition. The interplay of aldosterone, MR, and the kidney plays an indispensable role in the precise control of total body sodium, potassium, and chloride. In humans, renal glomeruli filter 150~180 L of plasma each day; however, 99% is reabsorbed by the renal tubules, leaving ~1% in the urine. The proximal tubule is responsible for the rough recovery of filtered NaCl, and the fine tuning of the NaCl reabsorption occurs at the distal nephron, the main site of action for aldosterone and MR in the tubules. In the principal cells of the collecting duct, aldosterone is responsible for the activation of MR because cortisol, which has a similar affinity for MR as aldosterone, is catalyzed to cortisone by 11β-hydroxysteroid dehydrogenase type 2. In the neighboring intercalated cells, activation of MR is regulated at the receptor level through phosphorylation at the ligand-binding domain [46,47]. This modification is regulated by angiotensin II (AngII) signaling involving the mammalian target of rapamycin and unc51-like kinase 1, regulating MR and the downstream target pendrin, a Cl^−^/HCO_3_^−^ exchanger in these cells [48,49,50,51]. In late distal convoluted tubules, NaCl reabsorption through thiazide-sensitive NaCl cotransporter (NCC) is regulated by mechanisms involving the ubiquitin ligase Kelch-like 3/Cullin-3 and Ser/Thr kinase with-no-lysine kinases [52,53], which is controlled by signaling that it does not require MR in these cells [52,54,55,56,57]. In addition, a recent study has shown that, in the late distal convoluted tubules, ENaC is likely regulated by glucocorticoids [58]. Therefore, intricate and coordinated actions of MR-dependent and -independent signaling optimize NaCl transport mechanisms in the distal nephron, thereby precisely controlling salt handling in the segment [52,59,60]. 

### 2.2. Kidney Disease and MR 

In addition to their roles in controlling fluid transport in the renal tubules, aldosterone and MR have been implicated in the progression of kidney injury. In the initial studies, DOCA was used to show the injurious effects of mineralocorticoids in the kidney. In 1943, Selye et al. reported that the combined administration of salt and DOCA causes malignant nephrosclerosis in rats [10]. In the kidney of that model, the authors described diverse pathological changes, such as glomerular capillary hyalinosis, tubular dilatation, and tissue fibrosis [10]. Aggravation of renal injury by DOCA has also been reported in other models of experimental nephritis and hypertension [61,62,63].

The development of angiotensin-converting enzyme inhibitors (ACEis) and AngII receptor blockers (ARBs) has highlighted AngII as an important mediator of CKD progression [64,65,66]. Nonetheless, several studies have also indicated the key role of aldosterone and MR in CKD progression among the components of the renin-angiotensin-aldosterone system. In an elegant study by Greene and colleagues [67], the rats were divided into four groups: sham-operated rats, rats that underwent 5/6 nephrectomy (remnant rats), remnant rats treated with enalapril and losartan, and enalapril- and losartan-treated remnant rats infused with aldosterone. As expected, enalapril and losartan markedly attenuated proteinuria, blood pressure elevation, and glomerulosclerosis. However, exogenous aldosterone infusion almost completely reversed the preventive effects of enalapril and losartan on proteinuria and glomerulosclerosis in the remnant rats, emphasizing the role of aldosterone and MR in the renal injury. Based on the previous study showing that MR mRNA is detectable in the glomeruli isolated by microdissection [68], the authors have discussed the possible contribution of direct cellular effects, in addition to blood pressure and other hemodynamic effects. Subsequently, the protective role of MR blockade in retarding glomerulosclerosis in the rat remnant kidney model has been shown by Fogo and colleagues [69]. In their study, the remnant kidney rats underwent renal biopsy at 8 weeks, and the rats were assigned to control, spironolactone, spironolactone plus non-specific antihypertensive agents, and spironolactone plus losartan groups. The assignment was based on the results of the biopsy; therefore, the degree of glomerulosclerosis in each group was similar at 8 weeks. At 12 weeks, the sclerosis index in the spironolactone group increased by 84%, as compared with a 157% increase in control rats. The protective effect of spironolactone was further enhanced by blood pressure control with the coadministration of non-specific antihypertensive agents or losartan. Intriguingly, the authors noted the regression of glomerulosclerosis in several rats in the spironolactone group. 

In more recent studies, the protective effects of MR blockade have been demonstrated in various models of hypertensive as well as non-hypertensive kidney diseases. Previously, we have demonstrated that MR blockade almost completely reverses massive proteinuria and prevents the development of glomerulosclerosis in mice lacking Rho-GDP dissociation inhibitor α (RhoGDIα) [70], which has later been proved to be a model of congenital nephrotic syndrome in humans [71,72]. MR blockade has also been shown to attenuate kidney injury in models of diabetic kidney disease [73,74], anti-glomerular basement membrane glomerulonephritis [75,76], and obstructive nephropathy [77,78,79]. Furthermore, MR blockade is shown to prevent the occurrence of acute kidney injury induced by ischemia/reperfusion (I/R) [80] and also prevent the transition of acute kidney injury to CKD [81,82]. Importantly, these models do not necessarily show blood pressure elevation or hyperaldosteronism. Mechanistically, a number of molecules and mechanisms have been identified that modulate MR activity (Figure 1), which can potentiate local MR signaling in the absence of a systemic increase in aldosterone [70,83,84,85,86,87]. Thus, these data indicate the diverse pathological consequences of MR overactivity, point to the intricate mechanisms leading to enhanced MR signaling, and highlight its essential role in the progression of kidney injury. 

### 2.3. Vascular Injury

One of the prominent features of the tissue damage mediated by MR is vasculopathy. Rocha et al. reported that thrombotic microangiopathy and fibrinoid necrosis observed in several hypertensive models (such as L-NAME/AngII and stroke-prone spontaneously hypertensive rats) are mediated by aldosterone and MR [88,89]. Although MR is most abundantly present in the distal tubules, the detection of MR outside the distal nephron in the human kidney has been demonstrated by immunohistochemical analysis [90] and more recently by single cell transcriptome analysis [91,92]. The creation and characterization of mice lacking MR in a cell-specific manner in the last decades have elucidated the role of MR in these non-classical targets in detail. Jaffe and colleagues created mice lacking MR in smooth muscle cells and demonstrated that vascular smooth muscle cell proliferation induced by aldosterone is attenuated in these mice [93]. Further exploration of the mechanisms in that model revealed the possible contribution of type 1 vascular endothelial growth factor receptor, and the authors demonstrated that in vivo blockade of this receptor prevents the vascular remodeling. Furthermore, other studies have also shown the role of MR in endothelial cells and in myeloid cells in promoting vascular inflammation and remodeling [94,95,96,97], highlighting the importance of the interaction of MR among different cell types in promoting tissue injury. 

### 2.4. Glomerular Damage

Another key feature of MR-mediated kidney injury is proteinuria and glomerular changes. In DOCA/salt rats, it has been reported that the damage to the podocytes, cells that cover the external surface of the glomerular basement membrane and serve as the filtration barrier by forming interdigitating foot processes, plays a key role in the progression of renal injury [98]. In that study, the authors performed detailed ultrastructural analysis in the DOCA/salt model at 6 weeks and inferred that podocyte injury and its detachment from the glomerular basement membrane result in the denudation of glomerular capillaries, promoting glomerulosclerosis. In our work, we have analyzed podocyte injury in uninephrectomized rats receiving aldosterone and salt (aldo/salt rats) and demonstrated that the expression of nephrin and podocin, the key components of the slit diaphragm of the podocyte foot process, was profoundly decreased from an early stage [99]. Conversely, a podocyte injury marker, desmin, was highly induced in podocytes in aldo/salt rats already at 2 weeks, and we confirmed the degenerative changes and the retraction of foot processes by electron microscopy. Moreover, in that model, heavy proteinuria, podocyte injury, and glomerulosclerosis were almost completely prevented by an MR antagonist, eplerenone, along with a partial but significant reduction in systolic blood pressure. 

Several other studies demonstrated that aldosterone and MR signaling trigger podocyte damage through multiple mechanisms both in vitro and in vivo [100,101,102,103] and that MR blockade provides podocyte protection in CKD models [104,105]. Importantly, the anti-proteinuric effects of MR antagonists can be seen under normal or even low circulating aldosterone levels [70,106,107]. Using a mouse model of congenital nephrotic syndrome (RhoGDIα knockout mice), we reported that extensive podocyte injury and heavy albuminuria were blocked by eplerenone, as mentioned earlier [70]. In that model, serum aldosterone levels were comparable to those of wild-type littermates. In search of the factors that modulate MR signaling, we found that the small GTPase Rac1, a regulator of diverse cellular processes including actin cytoskeletal organization, superoxide generation, and trafficking of nuclear transcription factors, promotes nuclear MR translocation and activation [70]. Indeed, we demonstrated that Rac1, but not RhoA, is hyper-activated in the kidneys of RhoGDIα knockout mice. Therefore, hyper-activation of Rac1 owing to RhoGDIα inactivation in podocytes triggers MR signaling, promoting podocyte injury and glomerulosclerosis. Consistently, a distinct study by Gee et al. has shown that the inhibition of Rac1-MR signaling mitigates the nephrotic phenotype in a model lacking RhoGDIα [72]. Furthermore, the pathophysiological importance of the aberrant Rac1 activity in podocytes has subsequently been demonstrated in a wide array of kidney disease models [108,109,110,111,112] and the associated pathway is now considered to be the key therapeutic target to prevent CKD progression [113,114]. 

Using a model of salt-sensitive rats, we further reported that salt loading triggers Rac1, which, in turn, induces glomerular podocyte injury in a model of salt-sensitive hypertension [115,116,117]. Moreover, accumulating data indicate that the pathogenic significance of the signaling interaction between Rac1 and MR may not necessarily be limited to glomerular diseases. For example, Barrera-Chimal et al. demonstrated that Rac1 activation is triggered by renal ischemia and that the genetic deletion of Rac1 and of MR in vascular smooth muscle limits the renal injury [80]. Rac1-mediated activation of MR has also been shown to be involved in cardiac injury and in increased salt sensitivity associated with chronic inflammation [118,119,120].

In addition to podocytes, endothelial cells and mesangial cells are also involved in MR-mediated glomerular damage. In the Munich Wistar Frömter rat, a CKD model with reduced nephron numbers, a non-steroidal MR antagonist, finerenone, reverses endothelial dysfunction through increased nitric oxide availability and the upregulation of superoxide dismutase activity [121]. It has also been demonstrated that aldosterone and salt promote damage to the glomerular endothelial glycocalyx, which is mediated by matrix metalloproteases, increasing the glomerular sieving coefficient [122]. In mesangial cells, aldosterone and MR promote cell proliferation and the accumulation of extracellular matrix [123,124,125,126,127]. In particular, studies have shown the role of plasminogen activator inhibitor-1 (PAI-1) in these cells [126,128]. In cultured mesangial cells and fibroblast cells, aldosterone and TGF-β synergistically increased PAI-1 mRNA and protein, which resulted in decreased extracellular matrix degradation [128]. In the rat remnant kidney model, spironolactone reduced PAI-1 expression in sclerotic glomeruli and fibrotic tubulointerstitium [69]. In addition, aldosterone/salt-induced glomerular injury was attenuated in PAI-1 knockout mice [129]. In histological analysis, glomerular and mesangial areas were increased to a greater extent in wild-type mice than in PAI-1 knockout mice after aldosterone/salt treatment. These data demonstrate the contribution of PAI-1 to aldosterone-induced glomerular injury [129].

### 2.5. Inflammation and Fibrosis

The other important feature of MR-mediated renal injury is inflammation and fibrosis. In a study by Blasi et al., inflammatory and fibrotic changes in uninephrectomized rats receiving aldosterone and salt were associated with the upregulation of osteopontin, monocyte chemoattractant protein-1, interleukin-6 (IL-6), and interleukin-1β (IL-1β), all of which were significantly or non-significantly attenuated in rats receiving eplerenone. Infiltration of ED1-positive macrophages and type III collagen content in the interstitium was also attenuated by eplerenone, demonstrating that MR blockade protects the kidney by preventing the proinflammatory and fibrotic responses. Although how MR induces renal inflammation and fibrosis is not fully elucidated, studies to date indicate that local MR signaling in various cell types, as well as signaling interaction among these cells, mediates the process [76,102,130,131,132,133,134,135]. Particularly, several lines of evidence indicate a central role of MR signaling in immune cells [76,131,136]. Barrera-Chimal et al. found that chronic renal injury and fibrosis after bilateral I/R in mice were reduced by the non-steroidal MR antagonist finerenone and by the selective deletion of MR in myeloid cells (MR^MyKO^) [131]. Interestingly, in the MR^MyKO^ model, CD206^+^, M2 macrophages tended to increase, whereas proinflammatory macrophages with high levels of Ly6 decreased, suggesting that the imbalance between M1 and M2 macrophages is important for chronic kidney injury and fibrosis. Further analysis identified that the inhibition of MR in macrophages acts through IL-4 receptor signaling to facilitate the switching of macrophage phenotype. The importance of dysregulated M1/M2 macrophage balance is also suggested in the aldosterone/salt model of renal injury [136]. In addition, a recent study demonstrated that renal fibrosis induced by aldosterone/salt and uninephrectomy is attenuated in mice lacking neutrophil gelatinase-associated lipocalin (NGAL) in macrophages, suggesting a possible downstream pathway of MR in these cells [137]. 

## 3. Renal Injury Induced by Aldosterone Excess in Humans

As described above, experimental studies have shown that MR induces kidney damage by multiple mechanisms that involve vasculopathy, glomerular injury, and tubulointerstitial fibrosis (Figure 1). Consistently, several lines of evidence indicate that autonomous aldosterone production in PA is associated with renal injury in humans. In 1976, Beevers et al. reviewed 136 cases of PA and reported that renal parenchymal diseases and vascular complications were not uncommon in those patients [138]. In an analysis of the Primary Aldosteronism Prevalence in Italy (PAPY) study, which included 64 patients with PA and 426 patients with essential hypertension, covariate-adjusted urinary albumin excretion, as assessed by 24-hour urine collection, was significantly higher in the former than in the latter [139]. Similarly, in the Japan PA study (JPAS), the prevalence of proteinuria was significantly higher in PA than in matched hypertensive patients [140]. As for the glomerular filtration rate (GFR), several studies suggest that PA is associated with hyperfiltration [141,142], which may mask the deterioration of kidney dysfunction, particularly at an early stage [140,143,144]. However, in a study by Reincke et al. that included 408 patients with PA in Germany, the percentage of subjects with an elevated serum creatinine level was significantly higher in PA than in the control group (29% vs. 10%) [145]. In a regression analysis, the independent predictors of reduced GFR were age, gender, low potassium, and high aldosterone levels [145]. In another study, the prevalence of CKD in PA was significantly increased from 8–16% to 28–37% after intervention with adrenalectomy or MR antagonists [143]. In that study, the independent predictors of decreasing estimated GFR (eGFR) after intervention were urinary albumin and serum potassium levels. Ogata et al. compared renal biopsies from 19 patients with PA with those from 22 autopsy cases with similar eGFR and histories of essential hypertension and reported that segmental glomerulosclerosis, interstitial fibrosis, and arteriolar hyalinization were significantly more pronounced in PA patients. Overall, these data are in line with experimental studies showing that aldosterone excess and MR signaling trigger a variety of structural alterations of the kidney [146]. 

## 4. Renoprotective Effects of MR Antagonists in CKD Patients

### 4.1. Steroidal MR Antagonists

There are multiple clinical studies that tested the effects of spironolactone in patients with CKD. In 2001, Chrysostomou et al. reported for the first time that spironolactone decreased urinary protein. They introduced 25 mg/day of spironolactone to eight renal disease patients with > 1 g/day of proteinuria despite treatment with enalapril, an ACEi. The authors reported that urinary protein decreased by 54%, from 3.8 ± 2.5 g/day before treatment to 1.8 ± 1.0 g/day [147]. In 2005, Schjoedt et al. conducted a double-blind, randomized crossover study to compare the effects of spironolactone (25 mg/day) and a matched placebo in 20 type 1 diabetic patients with overt albuminuria (eGFR > 30 mL/min/1.73 m^2^) despite antihypertensive treatment, including the renin-angiotensin system (RAS) inhibitors. The authors found that urinary albumin after 2 months decreased from 831 mg/day (95% CI, 626–1106) on placebo to 584 mg/day (95% CI, 441–829) on spironolactone treatment [148]. The same group tested the effects of spironolactone in 20 diabetic kidney disease patients (both type 1 and type 2 diabetes) with nephrotic range proteinuria and reported that the mean urinary albumin levels were reduced by 32% by spironolactone after 2 months [149]. 

In 2006, Bianchi et al. evaluated the effects of 25 mg/day of spironolactone in 165 CKD patients due to chronic glomerulonephritis with proteinuria (urinary protein-to-creatinine ratio; UPCR > 1 g/gCr) already treated with ACEi/ARB [150] (Table 1). In that prospective, open-label, randomized study, UPCR after 1 year decreased from a mean of 2.1 g/gCr to 0.89 g/gCr in patients receiving spironolactone, whereas UPCR remained unchanged in the control group. Furthermore, although eGFR was not statistically different between the two groups after 1 year of intervention, a monthly decline in eGFR was lower in the spironolactone group than in the control group at the end of treatment. Blood pressure significantly decreased in patients treated with spironolactone (132.9 ± 0.8/78.5 ± 0.5 to 126.9 ± 0.8/75.6 ± 0.5 mmHg) compared with the control group (131.6 ± 0.6/78.1 ± 0.4 to 130.2 ± 0.6/77.3 ± 0.5 mmHg). Serum K levels in the spironolactone-treated group (5.0 ± 0.1 mEq/L) were significantly higher than those in the control group (4.3 ± 0.1 mEq/L) after 1 year. Medhi et al. performed a randomized, controlled trial (RCT) that included 80 hypertensive patients with a urinary albumin-to-creatinine ratio (UACR) > 300 mg/gCr and diabetes mellitus who were treated with lisinopril (80 mg/day) [151]. Patients were randomly assigned to placebo, losartan 100 mg/day, or spironolactone 25 mg/day. Although blood pressure levels after 48 weeks did not differ among the groups, the percentage changes in UACR were −24.6% (95% CI, −54.8 to +25.9%) in the placebo group, −38.2% (−59.3 to −5.9%) in the losartan group, and −51.6% (−70.2 to −21.4%) in the spironolactone group. In sum, multiple studies have consistently demonstrated that spironolactone reduces proteinuria, although those studies were not powered to detect the differences in GFR changes. It also needs to be noted that meta-analyses that included those studies have reported a significant increase in the incidence of hyperkalemia with the use of spironolactone [152,153]. Recently, in the AMBER trial, a multicenter, randomized, double-blind study of the novel potassium binder, patiromer, in patients with treatment-resistant hypertension and an eGFR ≤ 45 mL/min/1.73 m^2^, showed a significantly higher rate of spironolactone continuation in the patiromer group than the placebo group at 12 weeks [154].

As for eplerenone, Epstein conducted a double-blind RCT that included 240 type 2 diabetic patients with UACR ≥ 50 mg/gCr [153]. After a run-in with enalapril, the participants were assigned to receive a placebo, eplerenone 50 mg, or eplerenone 100 mg. The use of amlodipine was allowed for blood pressure control after 4 weeks. The percentage changes in UACR at week 12 from baseline and the incidence of hyperkalemia were the primary endpoints. In that study, although there were no significant differences in blood pressure changes among the groups, the change in UACR in the placebo group was −7.4%, whereas UACR was reduced by −41.0% and −48.7% in the eplerenone 50 mg and 100 mg groups, respectively (*p* < 0.001 vs. placebo), at 12 weeks. There were no significant differences in the incidence of sustained hyperkalemia (serum K levels > 5.5 mEq/L on two consecutive occasions) or severe hyperkalemia (serum K levels ≥ 6.0 mEq/L) between the eplerenone and the placebo groups. 

In non-diabetic CKD patients, the Eplerenone Combination Versus Conventional Agents to Lower Blood Pressure on Urinary Antialbuminuric Treatment Effect (EVALUATE) study was conducted to determine the anti-albuminuric effect of eplerenone [155]. In that double-blind RCT trial, 314 non-diabetic CKD patients with albuminuria (UACR 30–599 mg/gCr) received either eplerenone 50 mg/day or a placebo. The primary endpoint, which was a percent change in UACR at 52 weeks from baseline, was significantly lower in the eplerenone group than in the placebo group (−17.3% vs. 10.3%; *p* = 0.02). In the safety analysis, there was no occurrence of hyperkalemia (>5.5 mmol/L) in either group. The decrease in UACR did not significantly correlate with baseline plasma aldosterone concentrations. Interestingly, however, when the participants were divided into tertiles according to salt intake in a sub-analysis [156], the eplerenone-treated group in the highest sodium excretion tertile, but not the middle and lowest tertiles, showed a significantly greater reduction in UACR than the placebo subjects in the same tertile, supporting the experimental data that salt intake triggers MR activation. 

In a subgroup analysis of the DAPA-CKD (Dapagliflozin and Prevention of Adverse outcomes in Chronic Kidney Disease) trial, a renal protection effect was achieved with or without spironolactone or eplerenone at baseline, and the incidence of hyperkalemia tended to be lower in patients receiving MR antagonists (hazard ratio, 0.87; 95% CI, 0.69–1.10) [157]. Moreover, in a recently published randomized open-label crossover study of eplerenone in combination with sodium-glucose cotransporter inhibitors (SGLT2is), the percent change in UACR at 4 weeks was significantly larger in the group receiving eplerenone 50 mg/day plus dapagliflozin 10 mg/day (−53%) than in the single agent groups (dapagliflozin alone, −19.6%; eplerenone alone, −33.7%) [158]. As a post-hoc exploratory end point, the study also found that the combination therapy group had a significantly lower incidence of hyperkalemia than the eplerenone alone group.

### 4.2. Esaxerenone

Esaxerenone has been developed as a highly selective non-steroidal MR antagonist and was approved in Japan in 2019 for the treatment of hypertension. The blood pressure-lowering effects of esaxerenone have been demonstrated in several clinical trials [159,160,161]. For example, in a double-blind, placebo-controlled RCT that included 403 patients with essential hypertension, the participants were randomly assigned to placebo, eplerenone, and different doses of esaxerenone [159]. In that study, the least mean changes in sitting systolic blood pressure were −7.0 mmHg in the placebo group, −17.4 mmHg in the eplerenone group, and −10.7 mmHg, −14.3 mmHg, and −20.6 mmHg in the esaxerenone 1.25 mg/day group, 2.5 mg/day group, and 5 mg/day group, respectively. In 24-hour ambulatory BP monitoring (24-h ABPM), the least mean changes in 24-h ABPM systolic blood pressure were 0.0 mmHg in the placebo group, −11.4 mmHg in the eplerenone group, and −5.9 mmHg, −8.5 mmHg, and −17.2 mmHg in the esaxerenone 1.25 mg/day group, 2.5 mg/day group, and 5 mg/day group, respectively.

Several lines of evidence have shown the anti-albuminuric effects of esaxerenone in addition to reducing blood pressure [160,162,163]. In a multicenter, single-arm, open-label phase 3 trial in 56 patients with type 2 diabetes, hypertension, and a UACR ≥ 300 mg/gCr [162], esaxerenone reduced UACR at 28 weeks by an average of 54.6% (95% CI, 46.9 to 61.3%) from baseline. In the ESAX-DN study [163], which included 449 type 2 diabetes mellitus patients with hypertension and albuminuria (UACR 45 to <300 mg/gCr) who were treated with ACEi or ARB, participants were randomized to receive either esaxerenone or a placebo for 52 weeks. The primary endpoint, which was the remission of albuminuria at the end of the treatment, was observed in 22% of the esaxerenone group and 4% of the placebo group (an absolute difference of 18%; *p* < 0.001) [163]. In the esaxerenone group, there was a significant reduction in blood pressure from baseline (−10/−5 mmHg) [163]. A post-hoc analysis of the study indicated that around 10% of the UACR-lowering effect of esaxerenone could be mediated by the reduction in systolic blood pressure [164]. 

In phase 3 studies of esaxerenone, the incidence of hyperkalemia with serum K ≥ 5.5 mEq/L was seen in 1.7% (21/1250) [165]. Risk factors for developing hyperkalemia include advanced CKD, diabetes, and older age, in addition to the use of RAS inhibitors [165]. Starting at a low dose (1.25 mg/day) may mitigate the risk of hyperkalemia in patients with these risk factors [160]. In a retrospective analysis, coadministration of SGLT2is with esaxereone was associated with a reduction in serum K levels without influencing the anti-proteinuric and anti-hypertensive effects of esaxerenone [166].

### 4.3. Finerenone

Finerenone is a novel non-steroidal MR antagonist that has been developed and approved for the treatment of diabetic kidney disease. In the Mineralocorticoid Receptor Antagonist Tolerability Study–Diabetic Nephropathy (ARTS-DN) study, which included 764 CKD patients with type 2 diabetes and albuminuria [167], the study participants were randomly assigned to receive either finerenone or a placebo. In that study, UACR at day 90 was significantly reduced in the finerenone group compared with the placebo group. Finerenone in Reducing Kidney Failure and Disease Progression in Diabetic Kidney Disease (FIDELIO-DKD), a multicenter, double-blind RCT, has provided the first clinical evidence that MR blockade improves kidney outcome in patients with diabetic kidney disease [168]. In that study, 5674 patients with type 2 diabetes and CKD (those with urinary albumin of 30 to <300 mg/gCr, an eGFR of 25 to <60 mL/min/1.73 m^2^, and diabetic retinopathy or urinary albumin of 300 to 5000 mg/gCr and an eGFR of 25 to <75 mL/min/1.73 m^2^) were randomized to receive finerenone or a placebo. After a median follow-up of 2.6 years, the primary endpoint composed of renal failure, an eGFR reduction of 40% or more, or renal-related death was significantly lower in the finerenone group than the placebo group (17.8% vs. 21.1%). UACR was also reduced by 31% at 4 months post-treatment compared to the placebo group, and the difference between the two groups persisted until the end of the trial. 

The FIGARO-DKD trial, a multicenter, randomized, double-blind study showing the cardioprotective effects of finerenone, was also published in 2021 [169]. The FIDELITY study, which combined the FIGARO-DKD and FIDELIO-DKD studies, demonstrated an average of 36% reduction in UACR at 4 months and a significant reduction in the combined renal outcome (renal failure, >57% eGFR reduction, and renal disease-related death) (HR 0.77, *p* = 0.0002) [170]. The benefit on cardiorenal outcome was observed irrespective of the use of SGLT2is, glycemic control, and duration of diabetes in the FIDELITY cohort [171,172]. 

As for the hemodynamic effects, the least squares mean difference in systolic office blood pressure between finerenone and a placebo was −2.7 mmHg over the course of the trial in the FIDELIO-DKD study [168], which seems relatively modest compared with other MR antagonists. However, modification of the antihypertensive treatment was permitted in that trial, which might have caused a conservative estimate in the group difference [168,173]. The data on 24-h ABPM in a subset of the ARTS-DN study showed that placebo-adjusted changes in 24-h ABPM systolic blood pressure were −8.3 to −11.2 mmHg at day 90 [174]. In the subgroup analysis according to the baseline office systolic blood pressure quartiles of the FIDELIO-DKD [173], the reduction in kidney outcome was consistently observed irrespective of baseline office systolic blood pressure levels, and 13.8% of the reduction in kidney outcome was explained by the change in office systolic blood pressure.

**Table 1 ijms-24-07719-t001:** Summary of major clinical studies evaluating the renoprotective effects of MR antagonists.

Frist Author, Study Name (year)	Design	Participants	MR Antagonists and Dosages	Length	Primary Endpoint	Results
Bianchi (2006) [150]	Open-label RCT	n = 165, chronic glomerulonephritis	Spironolactone 25 mg/day	12 months	Urinary protein-to-creatinine ratio	Spironolactone, 2.1 ± 0.08 to 0.89 ± 0.06 g/gCr (*p* < 0.001 from baseline); conventional therapy, 2.0 ± 0.07 to 2.11 ± 0.08 g/gCr
Epstein (2006) [153]	Double-blind RCT	n = 240, type 2 DM with albuminuria	Eplerenone 50 mg/day or 100 mg/day	12 weeks	Change in the urinary albumin to creatinine ratio (UACR) from baseline (%)	Eplerenone 50 mg/day, −41.0% from baseline; eplerenone 100 mg/day, −48.4% from baseline; placebo, −7.4% from baseline (both *p* < 0.001 vs. placebo)
Ando, EVALUATE (2014) [155]	Double-blind RCT	n = 314, non-diabetic CKD with albuminuria	Eplerenone 50 mg/day	52 weeks	Change in the UACR from baseline (%)	Eplerenone 50 mg/day, −17.3% (95%CI: −33.54 to −0.94%) from baseline; placebo, +10.3% (95%CI: −6.75 to 22.3%) from baseline (*p* = 0.02 vs. placebo)
Ito, ESAX-DN (2020) [163]	Double-blind RCT	n = 449, type 2 DM with hypertension and albuminuria	Esaxerenone 1.25–2.5 mg/day	52 weeks	UACR remission rate *	Remission rate was 22% in the esaxerenone group and 4% in the placebo group (*p* < 0.001 vs. placebo)
Bakris, ARTS-DN (2015) [167]	Double-blind RCT	n = 764, type 2 DM and CKD with albuminuria	Finerenone 1.25–20 mg/day	90 days	Ratio of UACR at day 90 vs. baseline	Placebo-corrected mean ratio of UACR at day 90 relative to baseline was 0.79 for finerenone 7.5 mg/day, 0.76 for 10 mg/day, 0.67 for 15 mg/day, and 0.62 for 20 mg/day
Bakris, FIDELIO-DKD (2020) [168]	Double-blind RCT	n = 5674, type 2 DM and CKD with albuminuria	Finerenone 10 or 20 mg/day	2.6 years	Composite of kidney failure, >57% decrease in eGFR from baseline, death from renal causes	Primary outcome occurred 17.8% in finerenone group and 21.1% in placebo group (hazard ratio, 0.82; 95%CI 0.73 to 0.93; *p* = 0.001)

The incidence of hyperkalemia-related discontinuation of the regimen was 2.3% in the finerenone group as compared with 0.9% in the placebo group in FIDELIO-DKD [168]. In the sub-analysis of the study, the incidence of hyperkalemia was significantly lower in SGLT2i-treated patients than in non-treated patients (hazard ratio, 0.45; 95% CI, 0.27–0.75) [175], suggesting that the combination with SGLT2is may mitigate the risk of hyperkalemia. 

### 4.4. Meta-Analysis and Real-World Evidence of MR Antagonists on Kidney Protection

A meta-analysis by Alexandrou et al. that included 31 RCTs published before January 2018 has reported a significant reduction in UACR of −25% and UPCR of −54% by MR antagonists compared with the placebo [176]. The incidence of hyperkalemia was 2.6-fold higher than the placebo [176]. The analysis included data on spironolactone, eplerenone, and the ARTS-DN study, but not on the FIDELIO-DKD study or esaxerenone. In a meta-analysis that included the studies of non-steroidal MR antagonists, the hazard ratio for the composite kidney outcomes (a 40% or 57% reduction in eGFR, renal death, and kidney failure) was 0.83 (95% confidence interval, 0.75–0.91) in the treatment group (based on six studies with finerenone) [177]. A network meta-analysis of 17 studies examining pre- and post-treatment UACR changes in diabetic patients with UACR > 30 mg/gCr with MR antagonists, SGLT2is, or their combination showed that the combined use of MR antagonists and SGLT2is is associated with low levels of albuminuria compared with MR antagonists, SGLT2is, or placebo alone [178]. In a single-center observational study of 3195 CKD patients with an eGFR of 10–60 mL/min/1.73 m^2^, the rate of initiation of renal replacement therapy was significantly lower in patients treated with MR antagonists than in those treated without MR antagonists [179], with a hazard ratio of 0.72, which also suggests that the use of MR antagonists is associated with improved kidney outcome.

The 2022 KDIGO guidelines recommend the induction of MR antagonists in diabetic patients with cardiorenal disease and albuminuria (UACR > 30 mg/gCr) despite adequate RAS inhibitor therapy if the patients have an eGFR > 25 mL/min/1.73 m^2^ and normal serum K concentrations (level of evidence 2A) [180]. The 2023 ADA Standards of Diabetes Care also recommended MR antagonist administration in diabetic CKD patients at risk for cardiac events and renal function decline (level of evidence A) [181]. 

### 4.5. Ongoing Clinical Studies and MR Antagonists in Development

The FIND-CKD trial was initiated in September 2021 and includes non-diabetic CKD patients with an eGFR of 25–90 mL/min/1.73 m^2^ and a UACR of ≥200 to ≤3500 mg/gCr [182]. The primary endpoint of this study is the change in eGFR from baseline to 32 months in the placebo and finerenone groups. The CONFIDENCE trial started in June 2022 to investigate the anti-albuminuric effects of empagliflozin in combination with finerenone [183]. The primary endpoint of the study is the change in UACR after 180 days of treatment with finerenone and empagliflozin, or with these drugs alone, in stage 2 or stage 3 CKD patients with type 2 diabetes mellitus and a UACR of ≥300 to <5000 mg/gCr [183].

There are also several novel non-steroidal MR antagonists that are under development, such as KBP-5074 (KBP Biosciences) [184] and apararenone (MT-3995, Mitsubishi Tanabe Pharma) [185]. In the BLOCK-CKD study, a phase 2, multicenter, randomized, double-blind study of KBP-5074 in stage 3b/4 CKD patients with poorly controlled hypertension [186], KBP-5074 significantly reduced systolic blood pressure by −7.0 mmHg (for 0.25 mg/day) and −10.2 mmHg (for 0.5 mg/day) after 84 days of treatment. No significant difference was noted in the median UACR, although there was a trend in reduction. As for apararenone, a phase 2 RCT was conducted in Japanese subjects with type 2 diabetes, an eGFR > 30 mL/min/1.73 m^2^, and a UACR of ≥50 to <300 mg/gCr [185]. Apararenone significantly reduced UACR at 24 weeks. At 52 weeks, the rate of remission in albuminuria was 5.1% for 2.5 mg, 30% for 5 mg, and 32.1% for 10 mg of apararenone. Apararenone also produced a reduction in systolic blood pressure in a dose-dependent manner [185]. 

## 5. Summary and Areas of Future Research

In this review article, we have summarized the current evidence obtained in basic as well as clinical studies concerning the deleterious effects of MR on kidney function and the renoprotective actions of MR blockade. Basic research has demonstrated that MR signaling can be potentiated not only by aldosterone but through other mechanisms, such as Rac1, cortisol, hyperglycemia, and oxidative stress, and that pathological MR overactivity results in diverse pathological consequences, including glomerulosclerosis, arteriolar hyalinosis, infiltration of inflammatory cells, and fibrotic changes in the tubulointerstitium as well as the perivasculature. As shown in Table 1, multiple clinical studies have also shown that MR blockade reduces proteinuria in CKD patients, when administered alone or in combination with ACEi/ARB. Moreover, the FIDELIO-DKD study with finerenone has provided the first evidence that the non-steroidal MR antagonist retards diabetic kidney disease progression [168]. Esaxerenone, another non-steroidal MR antagonist, is shown to reduce albuminuria in addition to its anti-hypertensive effect [163]. Several other non-steroidal MR antagonists are currently under development, which is worthy of attention. 

To optimize the clinical benefit of MR blockade in CKD patients, there are several important areas that need further investigation. Although several lines of evidence indicate that the combined treatment of SGLT2 and MR antagonists produces a profound reduction in albuminuria, it remains to be fully established whether the co-administration of these agents can additively retard the decline in GFR. A series of large clinical studies have demonstrated that finerenone is protective in diabetic kidney disease patients with albuminuria; whether it provides renoprotection in non-diabetic CKD patients with proteinuria merits further investigation. The long-term benefit of esaxerenone on kidney function also merits future evaluation, although previous studies did show that it reduced albuminuria, a proposed surrogate endpoint for kidney disease progression [187]. In addition, whether MR blockade attenuates I/R injury in kidney transplantation is an interesting subject to be pursued. Several ongoing studies may provide further information on these issues [188,189,190]. 

We would also like to note several key areas that would require future biomedical research regarding MR. First, the exact mechanisms by which MR blockade confers renoprotection are not entirely clear. The dissection of key downstream targets mediating kidney injury can open new opportunities to develop a specific inhibitor with fewer side effects. In relation to this point, to what extent the observed protective effects of MR blockade in clinical trials are attributable to hemodynamic and non-hemodynamic effects, respectively, deserves further examination. Although several studies have addressed this issue using office and sitting blood pressure, the analysis using 24 h blood pressure, as well as dipping phenotypes [191], may provide further insights. Second, emerging evidence points to the pathological effects of excessive sodium accumulation outside the extracellular fluids [52,192]. It would be of interest to determine whether the signaling mediated by tissue sodium is associated with local MR activity and whether the resultant pathologies are blocked by selective MR inhibition in these tissues. Finally, given that MR signaling can be modulated through multiple mechanisms involving steroidal and non-steroidal ligands, the identification of novel surrogate markers for MR signaling in the kidney can be useful in guiding treatment. Approaches to identify candidate markers may include the analysis of urinary nucleic acids and extracellular vesicles and their comprehensive profiling [193,194,195,196,197], which is worth exploring in future studies. 

## Figures and Tables

**Figure 1 ijms-24-07719-f001:**
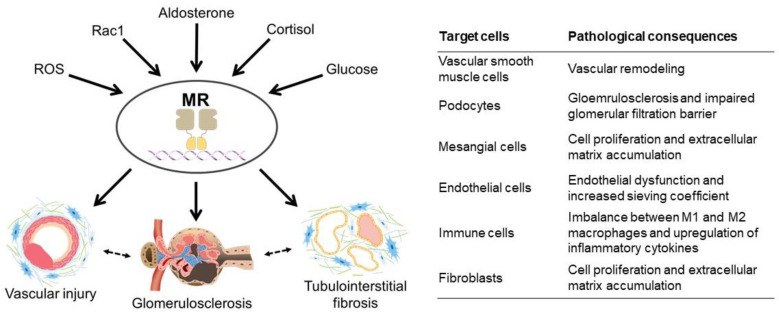
Mechanisms that trigger MR overactivity and the diverse pathological consequences. ROS, reactive oxygen species; MR, mineralocorticoid receptor.

## Data Availability

The data used in this article are sourced from materials mentioned in the References section.

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
