# Peer review of "Mineralocorticoid Receptor Antagonists for Preventing Chronic Kidney Disease Progression: Current Evidence and Future Challenges"

_ijms, 2023, doi:10.3390/ijms24097719_

Round 1

Reviewer 1 Report

Thank you for the opportunity to review this manuscript. Fujii and Shibata provide a comprehensive narrative review of mineralocorticoid receptor antagonists (MRAs) in the context of chronic kidney disease.

The authors provide an excellent overview of key historical events in the discovery and development of MRAs, both from a basic science and clinical perspective.

The authors provide adequate coverage of both steroidal and non-steroidal MRAs.

An area that is under much investigation, as the authors point out, is the use of MRAs together with SGLT-2 inhibitors. It may be worth mentioning the article "The Kidney Protective Effects of the Sodium-Glucose Cotransporter-2 Inhibitor, Dapagliflozin, Are Present in Patients With CKD Treated With Mineralocorticoid Receptor Antagonists" (DOI: 10.1016/j.ekir.2021.12.013).

A limitation on the clinical use of MRAs is hyperkalaemia, as mentioned in the manuscript. You could mention a study that has used potassium binders such as the AMBER trial "Patiromer and Spironolactone in Resistant Hypertension and Advanced CKD: Analysis of the Randomized AMBER Trial" (DOI: 10.34067/KID.0006782020).

In Figure 1, for completeness please define the abbreviations ROS and MR.

Author Response

An area that is under much investigation, as the authors point out, is the use of MRAs together with SGLT-2 inhibitors. It may be worth mentioning the article "The Kidney Protective Effects of the Sodium-Glucose Cotransporter-2 Inhibitor, Dapagliflozin, Are Present in Patients With CKD Treated With Mineralocorticoid Receptor Antagonists" (DOI: 10.1016/j.ekir.2021.12.013).

According to your suggestion, we cited the articled and mentioned in the main text (page 8, 2nd paragraph).

A limitation on the clinical use of MRAs is hyperkalaemia, as mentioned in the manuscript. You could mention a study that has used potassium binders such as the AMBER trial "Patiromer and Spironolactone in Resistant Hypertension and Advanced CKD: Analysis of the Randomized AMBER Trial" (DOI: 10.34067/KID.0006782020).

We appreciate the suggestion. We referred to the AMBER study in the revised version of the manuscript (page 7, 2nd paragraph).

In Figure 1, for completeness please define the abbreviations ROS and MR.

We defined the abbreviations.

Reviewer 2 Report

Dear Authors,

Your paper is very professionally written. Could the paper be improved? Yes, I recommend some improvements:

·         An uncommon critique is that your excellent text should become improved by the deletion of redundant acronyms. See my remarks 1, 2, 3, 4, 5, 10 below.

·         In Ch. 1 you better start the final two sentences (beginning with ‘In the following …’) at a new line.

·         In Ch. 2 I suggest inserting at least 4 sections. The text is too long. Suggestions: MR Blockades (e.g. after line 6 at page 3); Vascular Injury (below Fig. 1); Glomerular damage (page 6 , 6 lines from below); Inflammation and Fibrosis (page 6, after line 2 from above).

·         Please insert a table to clarify and summarize details the illustrative Figure 1 gives too little information of your results for the expert reader.

Details and discussions:

1.       Page 3, line 5 from below, you say ‘…, and also prevent the transition of AKI to CKD…’. I suggest modifying this in ‘…, and also prevent its transition to CKD…’.
Because you only have one usage of AKI. You better delete the acronym AKI in Page 3, line 6 from below.

2.       Page 4, line 14 from below:  the acronym VSMC in ‘vascular smooth muscle cell (VSMC}’ is useless because you do not use it in the sequel of your text.

3.       Page 4, line 11 from below :  the acronym VEGFR1 is also useless because you do not use it in the sequel of your text.

4.       Page 6, line 7 from above :  the acronym MCP-1 is also useless because you do not use it in the sequel of your text.

5.       Page 8, line 2 from above:  the acronym EVALUATE is also useless because you do not use it in the sequel of your text.

6.       Various pages: you use ‘SGLT2i’, ‘SGLT2is’, and ‘SGLT2’ is this inconsistent? Or all wrong, should the acronym  everywhere be ‘SGLT2I’ with a capital?

7.       At page 8 you have at least 3 times missing the determiner ‘the’ before ‘placebo’ and ‘ eplernone’ (2 x).

8.       Page 10, line 1 below ‘4.5.Ongoing…’:  the sentence should start with a determiner: ‘The FIND-CKD trial was initiated …’.

9.       Page 10, line 4 below ‘4.5.Ongoing…:’ the sentence should also start with a determiner: ‘The CONFIDENCE …’.

10.   Page 10, 7 lines lower, appears an unclear part of a sentence: ‘KBP-5074 (KBP Biosciences) and apararenone (MT-3995, Mitsubishi Tanabe Pharma).” Could you please explain the acronym, or is it redundant? And could you include a reference?

11.   The references list is not correct: inconsistent layout (with and without capitals, missing DOI numbers, missing page numbers, etc.).

Author Response

In Ch. 1 you better start the final two sentences (beginning with ‘In the following …’) at a new line.

We revised the manuscript according to your suggestion.

In Ch. 2 I suggest inserting at least 4 sections. The text is too long. Suggestions: MR Blockades (e.g. after line 6 at page 3); Vascular Injury (below Fig. 1); Glomerular damage (page 6 , 6 lines from below); Inflammation and Fibrosis (page 6, after line 2 from above).

Following your suggestion, we divided Ch. 2 into different sections (Electrolyte handling in renal tubules, kidney disease and MR, vascular injury, glomerular damage, inflammation and fibrosis).

Please insert a table to clarify and summarize details the illustrative Figure 1 gives too little information of your results for the expert reader.

We appreciate your suggestion. We inserted a table that summarizes the finding in Figure 1.

Details and discussions:

  1. Page 3, line 5 from below, you say ‘…, and also prevent the transition of AKI to CKD…’. I suggest modifying this in ‘…, and also prevent its transition to CKD…’.

Because you only have one usage of AKI. You better delete the acronym AKI in Page 3, line 6 from below.

  1. Page 4, line 14 from below: the acronym VSMC in ‘vascular smooth muscle cell (VSMC}’ is useless because you do not use it in the sequel of your text.
  2. Page 4, line 11 from below : the acronym VEGFR1 is also useless because you do not use it in the sequel of your text.
  3. Page 6, line 7 from above : the acronym MCP-1 is also useless because you do not use it in the sequel of your text.

We corrected the above.

  1. Page 8, line 2 from above: the acronym EVALUATE is also useless because you do not use it in the sequel of your text.

We consider it is necessary to put the information because “EVALUATE” is the name of the clinical study.

  1. Various pages: you use ‘SGLT2i’, ‘SGLT2is’, and ‘SGLT2’ is this inconsistent? Or all wrong, should the acronym everywhere be ‘SGLT2I’ with a capital?

Thank you for pointing out. We used the words “SGLT2is” through the text in the revised manuscript.

  1. At page 8 you have at least 3 times missing the determiner ‘the’ before ‘placebo’ and ‘ eplerenone’ (2 x).
  2. Page 10, line 1 below ‘4.5.Ongoing…’: the sentence should start with a determiner: ‘The FIND-CKD trial was initiated …’.
  3. Page 10, line 4 below ‘4.5.Ongoing…:’ the sentence should also start with a determiner: ‘The CONFIDENCE …’.

We corrected the text according to your suggestion.

  1. Page 10, 7 lines lower, appears an unclear part of a sentence: ‘KBP-5074 (KBP Biosciences) and apararenone (MT-3995, Mitsubishi Tanabe Pharma).” Could you please explain the acronym, or is it redundant? And could you include a reference?

“KBP-5074” and “MT-3995” are the code of the compounds defined by the company. We added the reference to the text.

  1. The references list is not correct: inconsistent layout (with and without capitals, missing DOI numbers, missing page numbers, etc.).

We checked and corrected the references.